# PKS 2155-304: A Case Study of Blazar Variability Power Spectrum at the Highest Energies and on the Longest Timescales

Arti Goyal 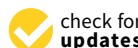

Astronomical Observatory of the Jagiellonian University, Orla 171, 30-244 Kraków, Poland; arti@oa.uj.edu.pl;
Tel.: +48-12-6238-620

**Abstract:** We present the results of our Power Spectral Density (PSD) analysis for the BL Lac object PKS 2155-304, utilizing the nightly-binned long-term light curve from the decade-long monitoring, as well as the minute-binned intra-night light curve from the High Energy Stereoscopic Survey (H.E.S.S.; >200 GeV). The source is unique for exhibiting the shortest flux-doubling timescale at Very High Energy (VHE) among its class and thus provides a rare opportunity to study the particle acceleration on the smallest spatial scales in blazar jets. The light curves are modeled in terms of the Continuous-Time Auto-Regressive Moving Average (CARMA) process. The combined long-term and intra-night PSD extends up to ~6 decades in the temporal frequency range; unprecedented at the TeV energies for a blazar source. Our systematic approach reveals that PKS 2155-304 shows, on average, a complex shape of variability power spectrum, with more variability power on longer timescales. The long-term variability is best modeled by the CARMA(2,1) process, while the intra-night variability is modeled by a CARMA(1,0) process. We note that the CARMA(1,0) process refers to an Ornstein–Uhlenbeck process where the power-law PSD slope (PSD varies as a function of variability frequency to the power of the negative slope) changes from two to zero, above a certain "characteristic/relaxation" timescale. Even though the derived power spectrum of the intra-night light curve did not reveal a flattening, we speculate such relaxation must occur on timescales longer than a few hours for the source.

**Keywords:** galaxies: active; BL Lacertae objects: individual (PKS 2155-304); variability

## 1. Introduction

Blazars, including Flat-Spectrum Radio Quasars (FSRQs) and BL Lac objects, are the most prolific class of GeV emitters in terms of apparent luminosity, constituting ~62% of point sources in the eight-year *Fermi*-Large Area Telescope (LAT) catalog of *Fermi*-associated sources (4FGL; [1]). Their characteristic two-peak Spectral Energy Distribution (SED) is believed to result from the nonthermal processes in a highly-magnetized relativistic nuclear jet. Within the leptonic scenario of blazar emission, both of these emission components are produced by the same population of ultra-relativistic charged particles (electron-positron ($e^{\pm}$) pairs) accelerated in the so-called *blazar-zone*. Even though the location and the main energy dissipation mechanism for the particle acceleration in this *blazar-zone* are extensively debated, the most favored scenarios include the formation of shocks and turbulence in the jet flow [2] or annihilation of magnetic field lines of opposite polarity at the magnetic reconnection sites [3] at ≤1 pc distances from the central Supermassive Black Hole (SMBH) [4]. The charged particles, accelerated up to TeV energies, produce low-energy emission (radio-to-optical, extending up to X-rays in the case of BL Lacobjects) via synchrotron processes, while the high-energy X-ray-to-$\gamma$-ray segment is most widely believed to be due to the Inverse-Comptonization (IC) of various circumnuclear photon

fields, produced either internally (Synchrotron Self-Compton; SSC) or externally by the accretion disk, the broad-line region, or the torus (External Compton; EC) to the outflow, by the jet electrons [5]. Alternatively, in the "hadronic" scenario, the high-energy emission continuum could also be generated via protons accelerated to ultra-high energies ($\geq EeV$), producing $\gamma$-rays via either direct synchrotron emission or meson decay and synchrotron emission of secondaries in proton–photon interactions, while the low-energy radiation is still dominated by synchrotron emission from $e^{\pm}$ pairs [6].

    Large total observed luminosities ($\sim 10^{47-48}$ ergs s$^{-1}$; [7]), coupled with a factor of few intensity changes on timescales as short as minutes at the highest energies, pose several challenges to the current understanding of blazar emission scenarios [8,9]. In this respect, the origin of short timescale flux variability and it's relation to larger amplitude variability on longer timescales is still widely debated. The variations on short timescales provide an additional challenge at the TeV$\gamma$-ray photon energies, as large jet bulk Lorentz factors ($\Gamma$) > 30–50 is needed to overcome photon opacity arguments [10,11], which are rather too extreme to be reconciled with the currently-favored models for the jet formation in blazar sources [12]. Moreover, the statistical properties of blazar light curves (from radio to $\gamma$-ray energies), in particular, the simple power-law shape of variability power spectral densities (PSDs; defined as P($\nu_k$)$\propto \nu_k^{-\beta}$, where $\nu_k$ is the temporal frequency, and $\beta \simeq$ 1–3 is the slope), indicate that the variability is generated by correlated noise-like processes on timescales ranging from decades to minutes (see, for a recent review, [13,14]). Specifically, the variations at synchrotron and IC energies seem to exhibit different statistical characteristics; one following a red/damped-noise process ($\beta \sim 2$), while the other following a pink/long-memory process ($\beta \sim 1$) (see [15] and the references therein).

    Despite the fact that blazars dominate the extra-galactic sky at High Energy (HE; >100 MeV) $\gamma$-rays, only a minority of them have been detected at Very High Energy (VHE; >100 GeV) $\gamma$-ray regime [1] . This is due to: (1) extreme photon deficiencies at the VHE energies due to power-law shapes of HE SEDs [16] and (2) the HE (>100 MeV) spectra of blazars showing absorption features that arise due to the interaction of the NHE (>100 GeV) photons with low energy photons from the Extragalactic Background Light (EBL; [17]), thus decaying into $e^{\pm}$ pairs [18–20]. Most of the time, due to the low operational duty cycles of the Cherenkov telescopes, the observations at the VHE are triggered due to flares from the lower energies [21]. Thus, the *unbiased* estimates of blazar emission and its variability on multiple timescales and in different flux states ("quiescence" vs. "flaring"), especially, up to the highest energies of the broad-band spectrum are difficult to obtain. In this regard, we point out that only due to the recent efforts of the First G-APD Cherenkov Telescope (FACT) [2] (starting from 2011; [22]) and the High–Altitude Water Cherenkov Gamma-Ray Observatory (HAWC) [3] (starting from 2014; [23]), a few other bright blazars are monitored on a daily basis without waiting for a trigger from other frequencies.

    PKS 2155-304 is an exception to this rule, which due to its relative proximity (redshift = 0.1160; [24]), favorable position in the sky, and high flux at VHEs, could be monitored on a daily basis with the High Energy Stereoscopic System (H.E.S.S.; 2004–2012; see [25,26] for the more recent multi-wavelength coverage). Moreover, this blazar underwent an exceptional flare at VHE energies on 28 July 2006 when the flux increased by more than a factor of $\sim$100 as compared to the average level, and a minute-like variability with flux-doubling time as short as 3-min was observed [9]. This timescale is shorter than a *light-crossing* timescale at the SMBH event horizon ($\sim$17 min for $2\times10^8$ M$_\odot$; [27]). Moreover, the slope, i.e., $\beta$, of the variability power spectrum (derived mostly using the methods in the *Fourier* domain), for the long-term (days to years timescales) and the intra-night (minutes to hours timescales), indicate $\beta \sim 1$ [25] and $\sim 2$ [9], respectively, indicating different statistical characteristics of the variability on long-term and intra-night timescales. We note that PSD estimation within the *Fourier*-domain approaches suffers from artifacts that arise due to an uneven sampling of the light curves, the finite

---

length of the light curve with underlying colored noise-type variability behavior, and discretization of the data (see [28] and the references therein). Techniques, such as the use of "window" function, as well as a linear interpolation of the time series, are used to minimize problems that essentially distort and introduce false data in the time series. Therefore, to avoid such issues in the derivation of PSDs, here we perform the power-spectral analysis of the long-term [4] and those publicly available [5] intra-night light curves using the *time*-domain method, in particular by applying the Continuous-time Auto-Regressive Moving Average (CARMA) model [6] by Kelly et al. [29]. Our analysis allowed us to construct the variability power spectrum at the highest energies of the electromagnetic spectrum and on the timescales extending up to ∼6 decades (years to minutes) for the first time for a blazar source.

Figure 1 presents the long-term (a) and intra-night (b) H.E.S.S. light curves of the blazar at energies >200 GeV. In Section 2, we outline the main features of the CARMA analysis. Results are given in Section 3, followed by a discussion and the conclusion in Section 4.

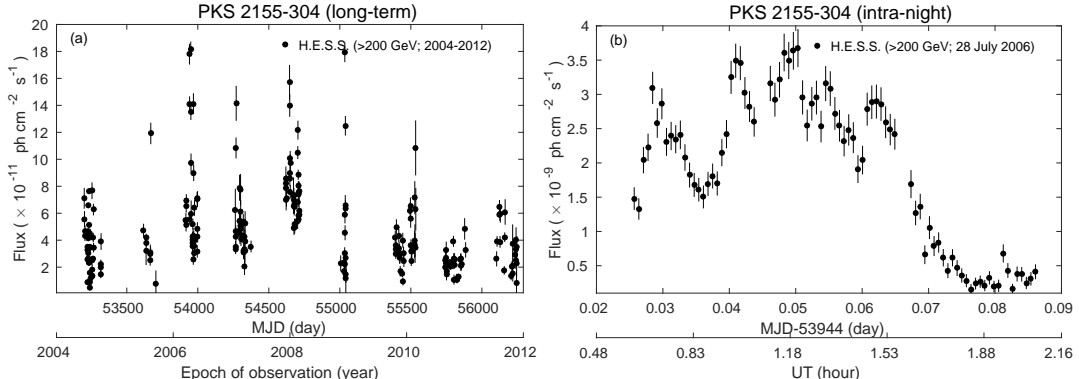

**Figure 1.** The long-term (**a**) and intra-night (**b**) H.E.S.S. light curves of the source.

## 2. Methodology: CARMA Modeling of the H.E.S.S. Light Curves

The reader is referred to Kelly et al. [29] for an in-depth discussion of the CARMA model. Here, we describe the main features very briefly. The measured time series $y(t)$ is approximated as a process defined to be the solution to the stochastic differential equation:

$$\frac{d^p y(t)}{dt^p} + \alpha_{p-1} \frac{d^{p-1} y(t)}{dt^{p-1}} + \cdots + \alpha_0 y(t)$$
$$= \beta_q \frac{d^q \epsilon(t)}{dt^q} + \beta_{q-1} \frac{d^{q-1} \epsilon(t)}{dt^{q-1}} + \cdots + \epsilon(t) \, , \tag{1}$$

where $\epsilon(t)$ is the Gaussian (by assumption) "input" white noise with zero mean and variance $\sigma^2$, the parameters $\alpha_0 \ldots \alpha_{p-1}$ are the autoregressive coefficients, and the parameters $\beta_1 \ldots \beta_q$ are the moving average coefficients.

We note that for a given light curve $y(t)$, one derives the probability distribution of the (stationary) CARMA($p, q$) process via Bayesian inference, and in this way, one calculates the corresponding power spectrum:

$$P(f) = \sigma^2 \left| \sum_{j=0}^{q} \beta_j \, (2\pi i f)^j \right|^2 \left| \sum_{k=0}^{p} \alpha_k \, (2\pi i f)^k \right|^{-2} , \tag{2}$$

along with the uncertainties. Kelly et al. [29] provided the adaptive Metropolis MCMC sampler routine to obtain the maximum-likelihood estimates. The quality of the fit was assessed by standardized

---

4   Data for the period 2004–2012 were obtained upon request from David Sanches.
5   https://www.mpi-hd.mpg.de/hfm/HESS/pages/publications/auxiliary/ApJL664_L71.html.
6   https://github.com/brandonckelly/carma_pack.

residuals: if the Gaussian CARMA model is correct, the residuals should form a Gaussian white noise sequence, for which the ACFis normally distributed with mean zero and variance $1/N$, where $N$ is the number of data points in the measured time series.

Here, the best-fit model parameters are obtained using the "corrected" Akaike Information Criterion (AICc; [30]). Finally, we note that the noise floor level in the derived PSD (Equation (2)) resulting from statistical fluctuations caused by measurement errors is calculated as:

$$P_{\text{stat}} = 2\,\Delta t\,\sigma_{\text{stat}}^2\,, \tag{3}$$

where $\Delta t$ is the sampling interval and $\sigma_{\text{stat}}^2 = \sum_{j=1}^{j=N} \Delta y(t_j)^2 / N$ is the mean variance of the measurement uncertainties in the flux values $y(t_j)$ in the observed light curve at times $t_j$.

## 3. Results

The flux distributions of blazar light curves can be modeled nonlinearly, in the sense that they often can be represented as $y(t) = \exp[l(t)]$, where $l(t)$ is a linear Gaussian time series (see, e.g., [25,31]). Hence, we have logarithmically transformed the light curves (Figure 1) and then modeled them as Gaussian CARMA$(p,q)$ processes. For each light curve, the minimum $(p,q)$ order was selected by minimizing the AICc values on the grid $p = 1, \ldots, 7$ and $q = 0, \ldots, p-1$ using the Markov chain Monte Carlo sampler with 10,000 iterations. We selected as the best-fit model the one produced by the pair of $(p,q)$ values having the lowest order within the range (see, [13], for the discussion).

The results of the CARMA model fitting are presented in Figure 2 (long-term) and Figure 3 (intra-night), respectively. Both of the analyzed light curves were well represented by the Gaussian CARMA process, as the residuals from the model fitting (to the measured time series) followed the expected normal distributions with the ACFs and the squared ACFs lying within $2\sigma$ intervals for most of the temporal lags. This is usually taken as a measure of the goodness-of-fit of the Gaussian CARMA model as any sample autocorrelations of the residuals should be independently and normally distributed with mean zero and standard deviation one. Note that because the long-term light curve is sparsely sampled, we estimated the noise floor level (Equation (3)) with either "mean" or "median" sampling intervals. Figure 4 presents the combined PSDs from long-term and intra-night monitoring.

1. The best-fit CARMA model for the long-term light curve turned out to be $p,q = 2,1$ (Figure 2). Even though the blazar was monitored fairly regularly with sampling intervals as close as one day, the large gaps in the monitoring due to seasonal gaps allowed us to cover timescales longer than $\sim$10 days up to $\sim$3000 days, above the "median" noise floor level.

2. The best-fit CARMA model for the intra-night light curve turned out to be $p,q = 1,0$ (Figure 3). This power spectrum is consistent with the Continuous-time Auto-Regressive (CAR(1)) process, according to the minimum AICc criterion adopted in this study.

3. The long-term and intra-night PSDs are plotted together in Figure 4. The combined PSD extended up to $\sim$6 decades, covering variability frequencies from 0.00033 day$^{-1}$ down to 720.0 day$^{-1}$. We note that the variability timescales shorter than a day and longer than a couple of hours could not be analyzed due to the daily sampling of the long-term light curve and relatively short duration of the intra-night light curve (the flare lasted for 1.45 h).

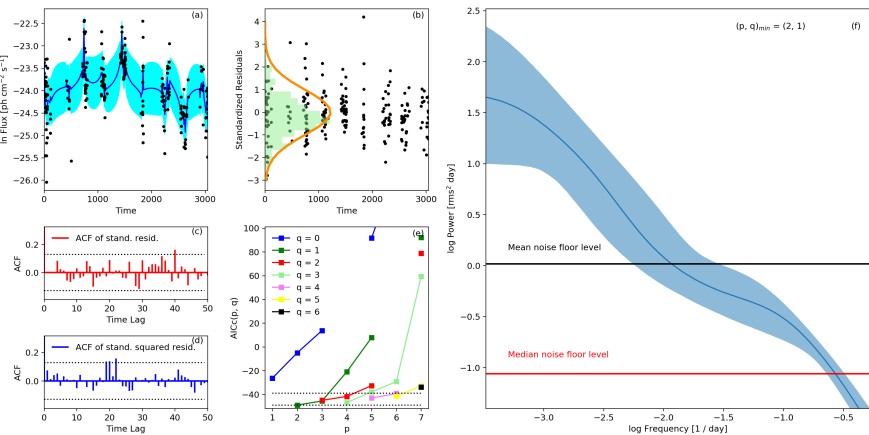

**Figure 2.** Results of CARMA modeling on the long-term light curve (Figure 1a). Panel (**a**) gives the logarithmically-transformed light curve along with the modeled values based on the best-fit Continuous-Time Auto-Regressive Moving Average (CARMA) process. Panel (**b**) shows the standardized residuals and their distribution compared with the expected normal distributions. Panel (**c**) gives the corresponding ACF compared with the 95% confidence regions, shown by dotted lines, for a white-noise process. Panel (**d**) gives the squared ACF compared with the 95% confidence regions, shown by dotted lines, for a white-noise process. Panel (**e**) gives the AICc values for different $(p, q)$ pairs (bands within the dotted lines refer to AICc values ranging from the lowest AICc to the lowest AICc + 10, which marks all the model sets, which are statistically indistinguishable from each other). Panel (**f**) shows the resulting PSD with the $2\sigma$ confidence region, as well as noise floor levels, $P_{stat}$, marked by horizontal black (=10.4 rms$^2$ day, corresponding to the mean sampling interval of 13 days) and red (=0.08 rms$^2$ day, corresponding to the median sampling interval of two days) lines, respectively. The best-fit CARMA model is obtained for $p, q = 2,1$.

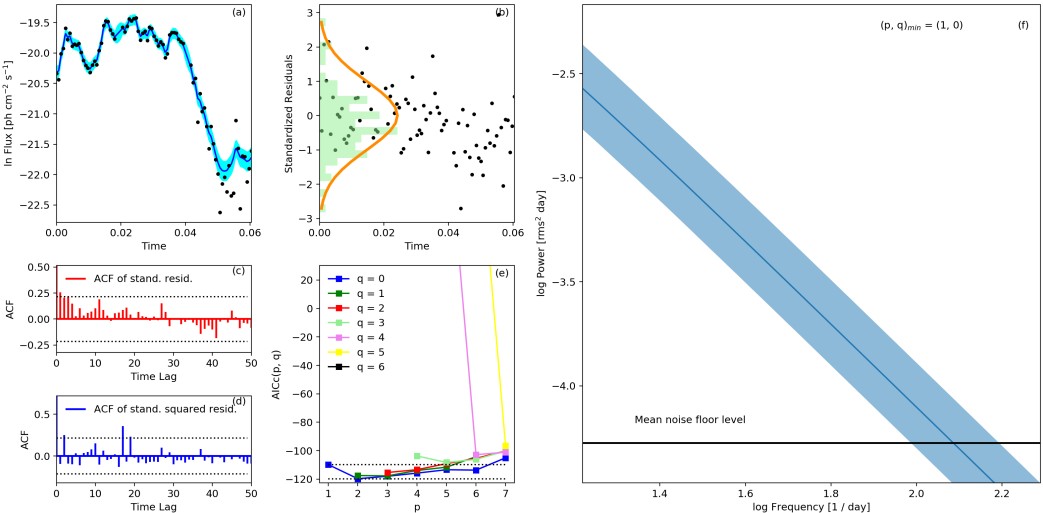

**Figure 3.** Results of CARMA modeling on the intra-night light curve (Figure 1b). The layout is the same as that of the Figure 2. $P_{stat}$ is 0.000053 rms$^2$ day, which corresponds to a mean (=median, in this case) sampling interval of 1.1 min. The best-fit CARMA model is obtained for $p, q = 1,0$.

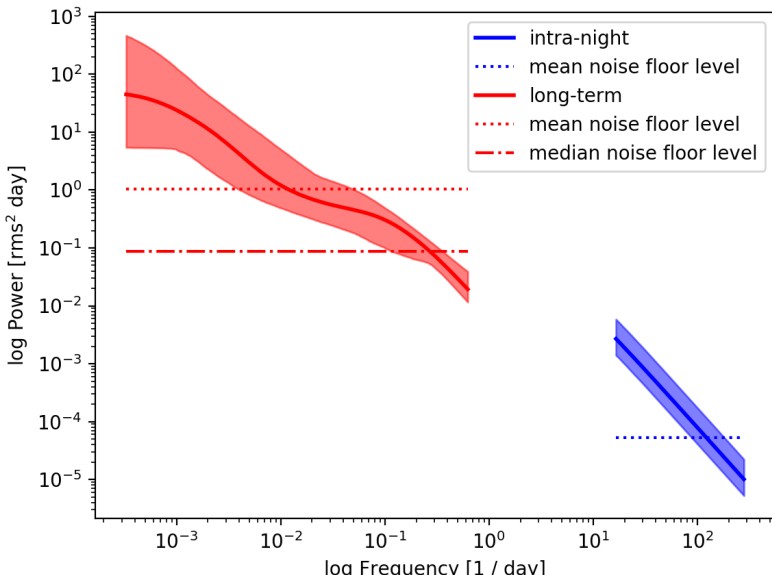

**Figure 4.** The combined long-term (shown in red color) and intra-night (shown in blue color) PSD from the H.E.S.S. monitoring, extending up to ~6 decades in the temporal frequency range. The shaded area corresponds to the $3\sigma$ confidence regions, and the dashed horizontal lines mark the $P_{\text{stat}}$, for the two datasets.

## 4. Discussions and Conclusions

PSDs characterized by $\beta \sim 1$–$2$ and the log-normal shapes of fluxes have been a common feature in galactic black binary systems where they are linked to underlying accretion processes [32]. Similarly, for blazar sources, the radio and $\gamma$-fluxes seem to follow a log-normal distribution [33,34], meaning that the variability is driven by a multiplicative process (such as the minijets-in-a-jet model; [35]), and not the additive ones such as superposition of many shocks or magnetic reconnection regions. The optical-to-VHE $\gamma$-ray fluxes of PKS2155-304 also follow a log-normal distribution [36], along with a flicker-noise-type variability characteristic for the HE and VHE $\gamma$-rays [25] and a red-noise-type variability characteristic at optical energies [37].

In [13,15], we interpreted that the different PSD slopes of variability could result if the broadband emission is generated in an extended, yet highly-turbulent jet. The statistically-different characteristics of synchrotron and IC frequencies was explained by hypothesizing that the synchrotron variability is driven by a single stochastic process operating on years to minutes timescales, while the IC variability is shaped by a linear superposition of two stochastic processes with different relaxation timescales (the former is the same as that of synchrotron variability, while the latter is related to small-scale plasma conditions). We speculated that the driver behind the former process could be related to the dissipation of the turbulent jet magnetic field supplied by an accretion flow and shaped by a combination of (global) MHD -timescales in a jet [38], while the additional one operating at $\gamma$-ray frequencies could be related to inhomogeneities in the local populations of soft photons available for the IC upscattering, leading to the "light-crossing timescale" relaxation of ~one day (for a jet with Doppler boosting factor $\delta \sim 30$).

Our results presented the variability PSD analysis, covering timescales ranging from $\geq 3000$ days down to 10 min at VHE $\gamma$-rays ($>200$ GeV) for the blazar PKS 2155-304. The CARMA modeling of the long-term light curve indicated a more complex shape of the variability process ($p, q = 2,1$) as compared to the intra-night light curve ($p, q = 1,0$), which is essentially a first-order Continuous-time Auto-Regressive (CAR(1)) model, also known as the Ornstein–Uhlenbeck process. For the CAR(1) model, the source variability was described as a damped random walk process, which can be described by the exponential covariance function $S(\Delta t) = \sigma^2 \exp(-|\Delta t / \tau|)$ defined by the amplitude $\sigma$ and the characteristic (relaxation) timescale $\tau$, shorter, for which the PSD slope is equal to two (strikingly

similar to PSD slopes obtained using the other methods; see Section 1). However, in our analysis, we did not identify a flattening of the intra-night PSD corresponding to a Gaussian (white) noise process, and due to the lack of coverage of variability frequencies $\sim$10 day$^{-1}$ to $\sim$0.1 day$^{-1}$, we speculate that such a relaxation must occur on timescales >few hours and <days. This implies a non-stationarity of the variability process on timescales >few hours. This is consistent with the log-normal distribution of fluxes from daily monitoring (see, [25,36]). We note that relaxation timescales around $\sim$days have been reported in the LAT light curves of a couple of other blazars, as well [31,39,40].

Finally, the close correspondence between the long-term and intra-night PSDs is remarkable and has been seen only in a couple of blazars, at HE $\gamma$-rays using the LAT monitoring, such as 3C 279 [40] and PKS 1510-089 [21]. At this point, we only mention the GeV emission results from IC scattering of seed photons that proceeds in general mainly in the Thompson regime, while in the case of the TeV emission, the Klein–Nishina effects become typically more relevant, which could be one of the factors shaping the amplitudes of the observed flux changes (see in this context, e.g., [41]).

**Funding:** This research was funded by the Polish National Science Centre (NCN) through the grant 2018/29/B/ST9/02298.

**Acknowledgments:** A.G. thanks David Sanchez (H.E.S.S.) for kindly providing the light curves in electronic form. We thank the referees for their careful reading of the manuscript and the constructive comments. Help and discussions on the CARMA analysis of the light curves with Volodymyr Marchenko are duly acknowledged.

**Conflicts of Interest:** The author declares no conflict of interest.

## Abbreviations

The following abbreviations are used in this manuscript:

PSD        Power Spectral Density
CARMA    Continuous-time Auto-Regressive Moving Average

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
