# Peer review of "PKS 2155-304: A Case Study of Blazar Variability Power Spectrum at the Highest Energies and on the Longest Timescales"

_galaxies, doi:10.3390/galaxies7030073_

Round 1
Reviewer 1 Report
The topic of the paper is very important as it applies a cutting edge time-domain method to possibly one of the best monitored sources at TeV energies and two highly significant datasets. And the results are very intriguing and seem to corroborate and strengthen the current understanding of TeV variability of PKS 2155-304. Therefore, it is important to publish these results.
However, there are some improvements required in the paper requested upon addressing which the paper can be recommended for acceptance. They are listed as follows with significant ones in the first 3 points.
1. It is emphasized that the PSD is valid for 6 decades, however 2 decades of timescales are not currently probed by observations as is clear from figure 4. And so in absence of data, this stance must be softened throughout the paper (detailed comments ahead)
2. While the method is detailed in Kelly et al., 2014, it would be very useful to write a few sentences on advantages of time-domain estimates over frequency domain ones (Fourier-based), in section 2. This would make the paper self-contained and therefore a lot more readable giving context to readers.
3. On line 99 I seek clarification on the following:
If y(t) is a CARMA(p,q) process, it is Gaussian (since white noise term is also Gaussian). Then I need to supply it with Log[Observed Fluxes] ; if observed fluxes are represented by l(t) then shouldn't it be, y(t) = Log[l(t)] as stated in line 100 by the author ? Perhaps this is merely a notational issue, but would be nice for the author to clarify. If not a notational issue, but a bug then it is very important to fix this.
4. In fig. (2) panel (e), why are there missing grid values ; for instance for q=0, p=4 is missing, q = 1, p = 6 is missing. Also, verify the units of y-axis in panel (f)
5. Also, in fig (2) AIC values would depend on the dataset ; can the author explain why we see the interesting increasing trends in AIC values with p ? This is opposite, to the decreasing trends in Kelly et al., 2014, and if there’s some intuitive reason, this will be a good insight.
6. Are the dotted lines in Fig. 2 panel (c,d,e) 95 % confidence levels – would it be possible to label it ? Given the size of the figures, I wouldn’t insist
7. It would be nice the author could remind the reader in a line or two, why the normal distribution of ACF and standard residuals results show that the Gaussian CARMA model fitting is successful.
8. It would be helpful if the author described the main results in the figure 3 for the intra-day in the caption which are of course, different from the long-term in the previous figure
9. Results section point 3.
The comparison of the long-term (2.5 decades) and intra-night (1.5 decades) is indeed very useful. They are two seperate ranges of timescales with results once again, potentially pointing to different regimes. However, it is important to state outright that there is a 2.5 decade gap that is not being analysed here. Else, it would be misleading to suggest that a continuous 6 decade model is available.
10. For figure 4 and the results : The long-term and intra-day clearly show different models i.e. different (p,q) values or different slopes. Could the author quantify that the normalisation is consistent with an extrapolation between the two without a break ? And indeed if there is a need of break, in absence of data at those timescales, one cannot constrain this. Alternately, if there's a physical justification for extrapolation of the power-law for long-term this should be explained adequately.
11. Line 126 – 127 : Indeed lognormality in the PDF is strongly suggestive of multiplicative processes ; and it is initmately tied to physically motivated models. However, from the current quality of data in general and thereby analyses, it is important to caveat this with the fact that there maybe alternate distributions which are neither lognormal or normal that might be consistent with data and indicate a different type of process (eg. superposition of power-laws, Rayleigh, etc). And especially so if you claim non-stationarity which shape of the PDF is not conserved.
12. Line 134 : extra 'at'
Reviewer 2 Report
Dear Author,
the paper is well written and concise. I only have a limited number of minor comments, which I give in order of appearance.
l.3: The acronym H.E.S.S. stands for "High Energy Stereoscopic System"
l.62: The acronym HAWC stand for "High Altitude Water Cherenkov Experiment"
l.112: Section 3 and 4 bear the same name. Please, change.
l.117 & Fig. 3e: The text says that the best fit CARMA model is p,q=1,0. However, from the plot (e) in Fig. 3 it seems that p,q=2,0 is a better fit. Please, explain.
l.132: Something seems to be wrong with the citation of "Goyal et al. 1315".
l. 147: There is a comma to much in (p,q,=1,0).
